# Identification and characterization of probiotics isolated from indigenous chicken (*Gallus domesticus*) of Nepal

Mohan Gupta[1☯], Roji Raut[2☯], Sulochana Manandhar[2], Ashok Chaudhary[2], Ujwal Shrestha[1], Saubhagya Dangol[1], Sudarshan G. C.[1,3], Keshab Raj Budha[4], Gaurab Karki[5,6], Sandra Díaz-Sánchez[7], Christian Gortazar[7], José de la Fuente[7,8], Pragun Rajbhandari[2], Prajwol Manandhar[2], Rajindra Napit[2,9], Dibesh Karmacharya[2,9,10]*

**1** Purbanchal University (PU), Bhatkekopul, Kathmandu, Nepal, **2** Center for Molecular Dynamics Nepal (CMDN), Kathmandu, Nepal, **3** Centre for Research and Interdisciplinarity (CRI), Rue Charles V, Paris, France, **4** SANN International College, Chabahil, Kathmandu, Nepal, **5** Kathmandu Research Institute for Biological Science, Lalitpur, Nepal, **6** University of Toledo, Toledo, Ohio, United States of America, **7** SaBio, Instituto de Investigación en Recursos Cinegéticos IREC-CSIC-UCLM-JCCM, Ciudad Real, Spain, **8** The Department of Veterinary Pathobiology, Center for Veterinary Health Sciences, Oklahoma State University, Stillwater, Oklahoma, United States of America, **9** BIOVAC Nepal, Banepa, Nepal, **10** The School of Biological Sciences, University of Queensland, Brisbane, Queensland, Australia

☯ These authors contributed equally to this work.
* dibesh@biovacnepal.com, dibesh@cmdn.org

**Data Availability Statement:** All sequence files are available from the https://www.ncbi.nlm.nih.gov/genbank/ database (accession numbers ON955508, ON955509, ON955510, ON955511).

## Abstract

### Background

Excessive and irrational use of antibiotics as growth promoters in poultry has been one of key factors contributing to increased emergence of antibiotics resistant bacteria. Several alternatives for antibiotic growth promoters are being sought, and the search for effective probiotics to be used as feed additives is amongst the promising ones. Our study aimed to isolate and test potential probiotics bacteria from cloacal swabs of various indigenous chicken (*Gallus domesticus*) breeds from rural outskirts of the Kathmandu valley (Nepal).

### Methods

Selective isolation of probiotics was conducted by micro-aerophilic enrichment of sample in MRS Broth at 37°C, followed by culturing on MRS agar supplemented with 5 g/L of $CaCO_3$. Isolated bacterial colonies producing transparent halo were selected as potential lactic acid bacteria (LAB), and tested for their antibacterial activity, phenotypic and biochemical characteristics, acidic yield, and tolerance to acid and bile.

### Results

A total of 90 potential LAB were isolated from cloacal samples collected from 41 free-ranging chickens of indigenous breeds. Of these, 52 LAB isolates (57%) showed variable antibacterial activity to at least one bacterial pathogen. Of 52 LAB, 46 isolates fulfilled phenotypic and biochemical criteria of *Lactobacillus* spp. Of these, 37 isolates produced varying percentage

**Funding:** Partial fund for conducting the laboratory experiments were received as academic thesis support from SANN International College, Purbanchal University (by the authors Mohan Gupta, Ujwal Shrestha, Saubhagya Dangol, and Sudarshan GC). Rest of the fund for conducting this study was obtained from institutions BIOVAC Nepal and Center for Molecular Dynamics Nepal (CMDN). - Roji Raut, Sulochana Manandhar, Ashok Chaudhary, Pragun Rajbhandari, and Prajwol Manandhar received salary from Center for Molecular Dynamics Nepal (CMDN). - Rajindra Napit, and Dibesh Karmacharya received salary from BIOVAC Nepal. The funders had no role in study design, data collection and analysis, decision to publish, or preparation of the manuscript.

**Competing interests:** The authors have declared that no competing interests exist.

yields of lactic acid, 27 isolates showed survival at pH 3.0, and 17 isolates showed survival tolerances in the presence of 0.3% and 0.5% bile salts for 24 hours. Phylogenetic analysis of 16S rDNA sequencing of LAB isolates fulfilling *in vitro* probiotics properties showed that 3 isolates had genetic identity of 99.38% with *Lactobacillus plantarum*, while one isolate was genetically similar (99.85%) with the clade of *L. reuteri*, *L. antri* and *L. panis*.

## Conclusion

Our study identified four *Lactobacillus* spp. strains having potential probiotics properties. Further investigations are needed to evaluate these isolates to be used as poultry probiotics feed supplement.

## Introduction

An increasing global population and food security needs have imposed a great pressure on poultry and livestock sectors to increase their production utilizing limited resources [1]. Consequently, sub-therapeutic doses of antibiotics are being used widely in poultry as animal growth promoters and for prophylaxis [2]. This is particularly true in low and middle income countries where poultry industries are playing an important role in their national economy [3, 4], and where regulations on the use of antimicrobials are often weak [5]. Such routine and irrational use of antimicrobials has increased the risk of emergence and spread of antimicrobial resistance (AMR) among poultry associated bacterial population, endangering both poultry and human health [6, 7].

Probiotics are both live microorganisms and their metabolites, which when taken orally, provide health benefits to the receiving host by preventing enteric diseases and/or enhancing health performance and productivity [8, 9]. There are numerous ongoing efforts to discover effective probiotics strains for poultry health with similar beneficial effects as antibiotic growth promoters (AGPs). A successful probiotics candidate must be non-pathogenic and produce desirable *in vivo* benefits while surviving hosts' gastro-intestinal environment, such as tolerance to acidic pH and high bile salt concentrations. Additionally, such probiotics must have a good adherence to the intestinal epithelium and symbiotic colonization with the natural gut microbiota [10]. Lactic acid bacteria (LAB) including different species and strains of *Lactobacillus* are one of the most common types of bacterial microorganisms fulfilling these characteristics, and hence are increasingly used as probiotics [11, 12].

Several formulations of probiotics based on different strains of bacterial and non-bacterial agents are widely used in poultry sectors in high income countries [1]. However, the use and particular production of probiotics in poultry sectors in developing countries, like Nepal, is relatively less pronounced. Compared to commercial formulations composed of non-native strains, the use of probiotics bacterial strains isolated from indigenous chicken (*Gallus domesticus*) breeds may offer unique host-specific advantages of improved gut adherence and survival. In this study, we isolated LAB strains from cloacal swabs of various indigenous chicken breeds from Nepal that survived on natural foraging (without any known exposure to antibiotics), and selected potential probiotics candidates based on *in vitro* challenge tests for survival in presence of varying bile concentrations and acidic pH, percentage yield of lactic acid and inhibition of selected bacterial pathogens- thereby identifying bacterial isolates with unique and improved probiotics characteristics.

## Materials and methods

Sampling was done after obtaining written consents from the farmers. The survey included questions on use of antibiotics and feed source. As this study deals with microorganism (bacteria), and that there is no direct handling of the animals, this study is exempt from any ethical considerations.

### Specimen collection and preparation

Free-ranging backyard chicken primarily belonging to various indigenous breeds reared in rural outskirt hills of the Kathmandu valley were selected for sampling (Fig 1, S1 Table). The selected birds lacked exposure to commercial and/or antibiotics supplemented feed. Cloacal samples (n = 41) were collected from each chicken using sterile swab sand preserved in cryovials containing 50% sterile glycerol. The samples were then transported in cold chain box to the BIOVAC laboratory in Kathmandu and stored at -20˚C.

### Selective isolation of LAB isolates

Glycerol stock of cloacal swabs was thawed in room temperature. After vortexing, 50 μl of each sample was inoculated into 5 ml of DeMan, Rogosa and Sharpe (MRS) Broth (HiMedia, India) with 0.5% calcium carbonate ($CaCO_3$). The inoculated broth was incubated at 37˚C for 72 hours in carbon-dioxide enriched environment using candle jar incubation method.

After incubation, 50 μL of inoculated broth was streaked on MRS agar (HiMedia, India) with 0.5% calcium carbonate. The agar plates were incubated at 37 ˚C for 24 hours in carbon-dioxide enriched environment. White to cream colored round colonies (3–4 mm) producing clear hydrolyzing halos were selected as potential LAB isolates. Multiple morphologically distinct colonies producing clear hydrolyzing zones per sample were selected as these could be potentially distinct LAB isolates. The selected colonies were further re-plated on 0.5% $CaCO_3$ supplemented MRS agar for pure culture isolation. The overnight MRS broth culture of each selected LAB isolates were prepared, and subjected to further *in vitro* screening and challenge tests as below.

### Test for anti-bacterial activity

The anti-bacterial activity of selected LAB isolates were tested against a panel of bacterial pathogens using agar well diffusion method [13]. Surface of Muller Hinton Agar (MHA) plates were lawn cultured with 0.5 $O.D_{630}$ adjusted bacterial suspension of six selected pathogens (*Salmonella enterica* spp., *Escherichia coli*, *Shigella* spp., *Klebsiella pneumonia*, *Citrobacter freundii*, and *Staphylococcus aureus*). After 15–20 minutes, six wells ($\geq$ 30 mm apart) were bored with the broader end of a sterile pipette tip on the inoculated MHA agar. A 100 μl of the overnight MRS broth culture of each selected LAB isolate (adjusted to 0.5 $O.D_{630}$) was then inoculated into the wells. The broth was allowed to diffuse into the agar for approximately an hour by incubating the covered plates at 4˚C in an upright position. After complete diffusion, the plates were incubated in an inverted position at 37˚C for 24 hours. Development of clear zone of any size around each well was noted as positive anti-bacterial activity, and sizes of inhibition zone were recorded [14].

### Phenotypic bacterial identification

Presumptive phenotypic bacterial identification of the selected LAB isolates demonstrating anti-bacterial activity was performed by using standard microbiological methods, which included Gram staining and biochemical tests for catalase, oxidase, sulfur and indole

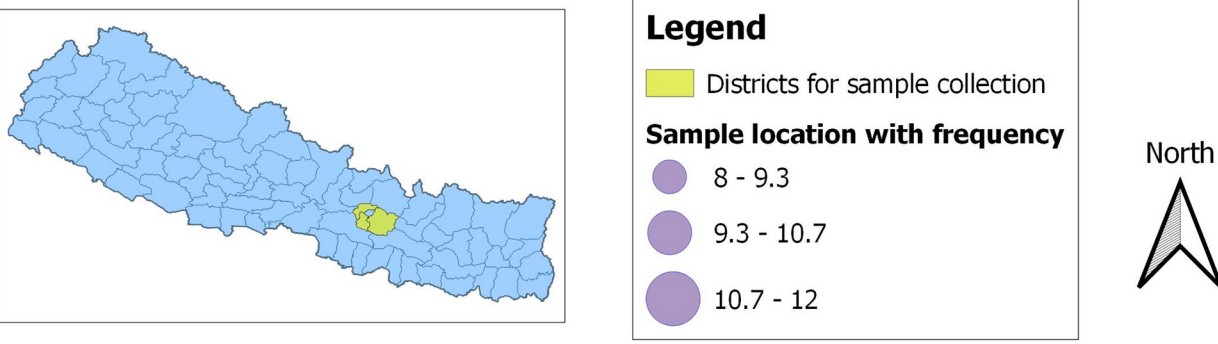

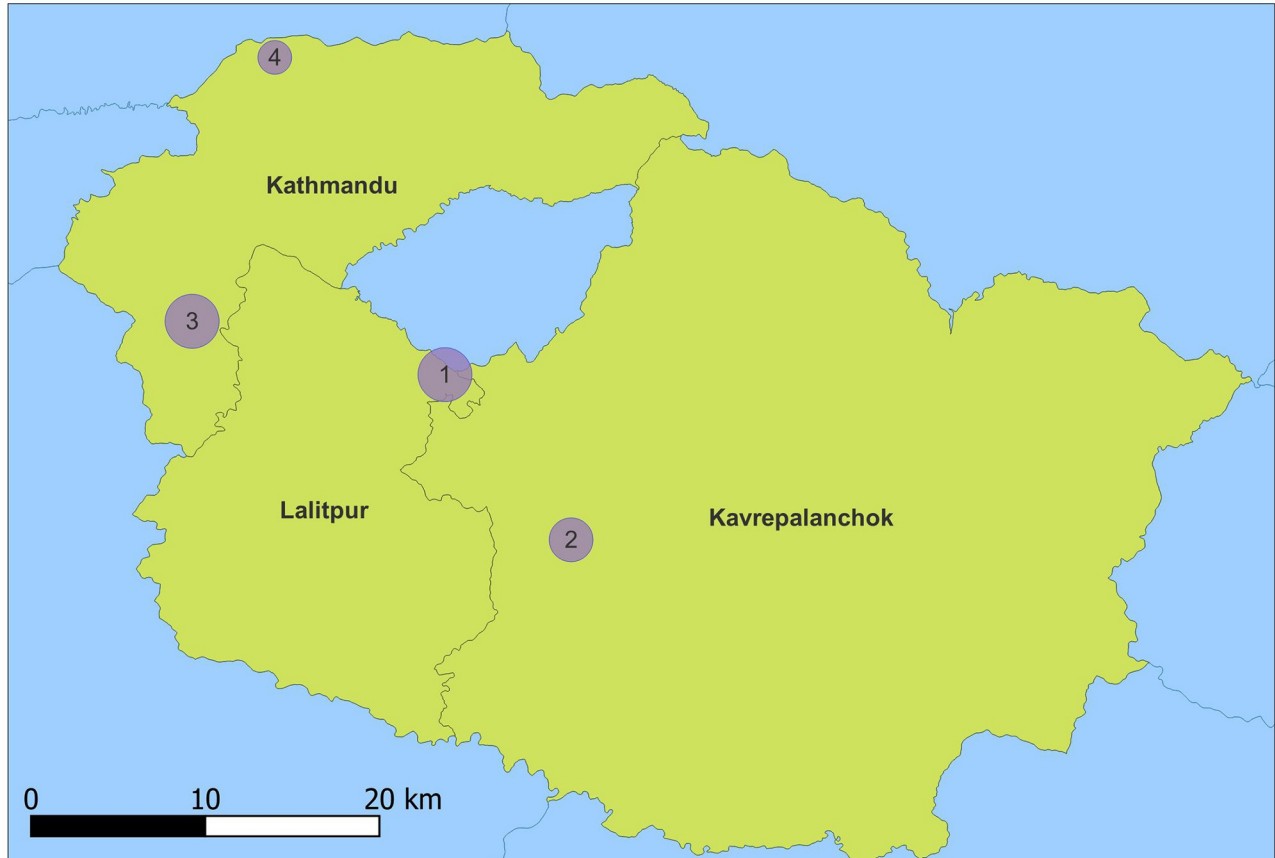

**Fig 1. The map of sampling districts (yellow colored) of Nepal showing the sampling locations indicated as numbered circles.** The sample locations are as follows: 1) Lakuribhanjyang, 2) Parthali- Bethankchowk, 3) Chalnakhel-Bosandanda and 4) Jhor- Dhakalchaur. The diameter of sampling locations are shown as varying proportionately with the number of samples collected from that location as indicated in the figure legend.

production [15]. The Gram positive rod shaped bacteria that were negative for catalase, oxidase, sulphur, and indole production were phenotypically identified as *Lactobacillus* spp.

## Test for production of total lactic acid

Lactic acid production by presumptive *Lactobacillus* spp. isolates was quantified. For this, overnight MRS broth culture of each *Lactobacillus* spp. was subjected to $10^{-1}$ dilution in sterile

distilled water. To this, 3 drops of 1% phenolphthalein was added, and was titrated with 0.1 N NaOH solution until a light pink colour was observed [16]. The volume of 0.1 N NaOH required to achieve neutralization was used to calculate the percentage yield of lactic acid using by the following calculation [17, 18]:

$$Percentage\ yield\ of\ lactic\ acid = \frac{Actual\ yield}{Theoretical\ yield} x100\%$$

The actual yield is the amount of lactic acid formed as a result of bacterial fermentation during overnight incubation, and theoretical yield is the maximum amount of lactic acid presumed to be produced by homo-fermentative LAB strains from available amount of fermentable sugar present in the MRS broth (i.e., 0.02 gm/ml).

## Test for acid tolerance

Overnight incubated MRS broth of potential *Lactobacillus* spp. testing positive for lactic acid production was centrifuged at 1,000 rpm for 10 minutes. Resulting bacterial pellet was re-suspended in 2 ml of sterile saline. One ml of this bacterial suspension was added to 9 ml of sterile artificial gastric juice media (0.2% NaCl, 0.35% pepsin, adjusted to pH 3.0). Another one ml of bacterial suspension was added to 9 ml of sterile MRS broth (adjusted to pH 7.0). The tubes were incubated at 37 ˚C for 3 hours and $O.D_{600}$ was measured for both the tubes after incubation [19]. For calculation, bacterial density at $O.D_{600}$ was empirically equated to the bacterial count in colony forming unit (CFU)/ml as per the growth curve calculation of *Lactobacillus plantarum* as described by Trabelsi et al. [20]. Acid tolerance ability of bacterial isolates was estimated using the following formula [19]:

$$Percent\ acid\ tolerance = \frac{CFU/ml\ at\ pH\ 7.0 - CFU/ml\ at\ pH\ 3.0}{CFU/ml\ at\ pH\ 7.0} x100\%$$

## Test for bile salt tolerance

A set of MRS broth with 0.2% sodium acetate, each supplemented with cattle bile salt (concentrations of 0.0%, 0.3%, 0.5%, and 1%) were prepared. Overnight activated MRS bacterial suspension (20 μl) of presumptive *Lactobacillus* spp. testing positive for lactic acid production was inoculated in the broth (180 μl) and incubated at 37 ˚C for 24 hours. After incubation, $O.D_{600}$ was measured using micro-plate absorbance reader (BioTek, USA). Bacterial absorbance density at $O.D_{600}$ was empirically equated to the bacterial count in CFU/ml as per the growth curve calculation of *Lactobacillus plantarum* as described by Trabelsi et al. [20]. The bile tolerance of the test isolates was calculated using the following formula [21]:

$$Bile\ tolerance = \frac{CFU/ml\ in\ 0.0\%\ bile - CFU/ml\ ingiven(0.3\%,\ 0.5\%\ or\ 1\%)\ bile}{CFU/ml\ in\ 0.0\%\ bile} x100\%$$

## Bacterial species identification by 16S rDNA sequencing

**Bacterial DNA extraction.**  Phenotypically identified presumptive *Lactobacillus* spp. yielding optimal *in vitro* probiotics characteristics were subjected to bacterial DNA extraction as per kit instructions (Zymo BIOMICS DNA Miniprep kit, USA). The 16S rDNA PCR was performed in a 10 μL reaction containing 1X KAPA HiFi-Hotstart Ready mix (KAPA, USA), 0.2 μM of each forward (5'CCTACGGGNGGCWGCAG3') and reverse primers

(5'GACTACHVGGGTATCTAATCC3') [22], and 1 μL of template DNA. PCR was conducted at 95 ˚C/15 minutes for initial denaturation, followed by 40 cycles of each 98 ˚C/30 s, 65 ˚C/30 s and 72 ˚C/30 s, with final extension at 72 ˚C/5 minutes. The amplicons (550bp) were excised from gel and purified using 1X Agencourt AMPure beads (Beckman Coulter, USA) and sequenced using both the forward and reverse primers on ABI-310 Genetic Analyzer (Applied Biosystems, USA). An M13 sequence was used as a positive control.

After initial quality assessment, the resulting DNA sequences were referenced in the NCBI database using BLAST (Basic Local Alignment Search Tool) [23]. For taxonomic assignment, phylogenetic analysis was performed by comparing sample sequences with a set of *Lactobacillus*16S rDNA sequences extracted from the NCBI GenBank. The best-fit nucleotide substitution model, K2+G, was selected based on the Bayesian Information Criterion (BIC) score in MEGA X v10.2.5 [24]. Maximum likelihood (ML) phylogeny was inferred using the selected model in MEGA X, with 1000 bootstrap replications. Phylogenetic tree annotation and visualization was performed using FigTree v1.4.4 [25].

## Results

A total of 41 cloacal samples were collected from various indigenous breeds of free-ranging backyard chicken from rural outskirt hills of the Kathmandu valley. A total of 90 potential LAB isolates were selected based on $CaCO_3$ hydrolysis in MRS agar.

Of 90 potential LAB isolates, 52 isolates (57%) showed anti-bacterial activity against at least one bacterial pathogen with zone of inhibition ranging from 7 to 18 mm (S2 Table). Majority (69%, 36/52) of LAB isolates inhibited *Salmonella* spp., followed by inhibition against *C. freundi* (53%, 28/52), *Shigella* spp. (44%, 23/52), *K. pneumonia* (30%, 16/52), *E. coli* (26%, 14/52) and *S. aureus* (13%, 7/52). Nine of these LAB isolates (17%) showed inhibition against all tested Enterobacterales bacteria.

Of 52 LAB isolates showing varying anti-bacterial activity, 46 fulfilled basic phenotypic criteria of *Lactobacillus* spp.; Gram-positive rods giving negative reactions for each sulphur, indole, catalase, and oxidase tests. The selected 46 potential *Lactobacillus* spp. isolates were further subjected to *in vitro* screening tests for probiotics properties. Of 46 isolates, 37 produced acid of varying percentage yields ranging from 38 to 81. Among these 37 isolates, 27 isolates showed survival at pH 3.0 with varying survival rates (1.2% to 62.1%). Tolerance to bile salt varied with the bile concentrations. 17 isolates had 0.3% and 0.5% bile salts tolerance, while 14 isolates survived in the presence of 1.0% bile salt. The resulting profile of 52 LAB isolates on *in vitro* screening tests for probiotics is given S2 Table.

Of 27 phenotypically identified *Lactobacillus* spp. exhibiting optimal *in vitro* probiotics properties, 3 isolates (26B, 28B, and 30B) showing anti-bacterial activity against the widest range of tested bacterial pathogens, and one (C4/36(4)) showing high bile tolerance (1.0%) were randomly selected for 16S rDNA sequencing. Based on phylogenetic analyses of 16S rDNA sequences of the isolates, 26B (GenBank accession number ON955508), 28B (GenBank accession number ON955509), and 30B (GenBank accession number ON955510) isolates clustered into the clade of *L. plantarum* with average p-distance of 0.0061, inferring a genetic similarity of 99.38% with *L. plantarum*. The isolate C4/36(4) (GenBank accession number ON955511), on the other hand, grouped into the cluster of *L. reuteri*, *L. antri*, and *L. panis* with average p-distance of 0.0014 and a genetic similarity of 99.85% with the given cluster (Fig 2).

## Discussion

In our study, we have isolated and identified potential LAB from 41 non-duplicate cloacal samples collected from a group of diverse backyard chicken breeds reared at a rural outskirt of the

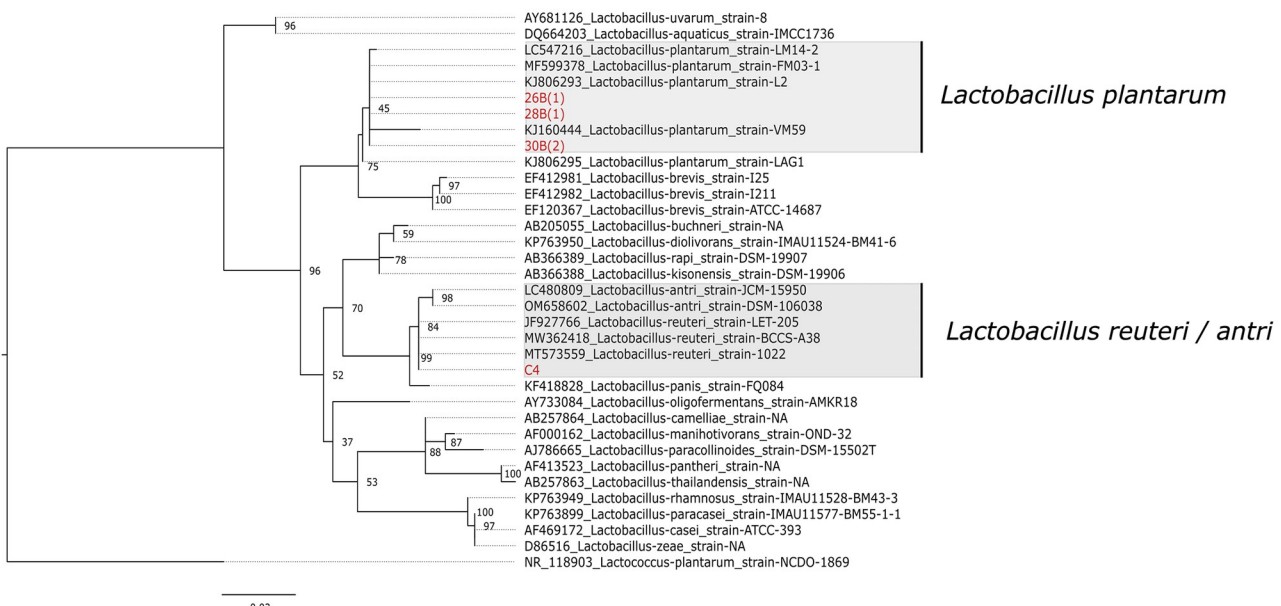

**Fig 2. Maximum likelihood tree based on 16S rDNA gene sequences of different *Lactobacillus* spp.** The isolates of the study are indicated in red font. *Lactococcus plantarum* was chosen as an out-group. The scale bar (0.03) shows the nucleotide substitution rate per site. Bootstrap probabilities as determined for 1000 replicates are shown at nodes. The isolate C4/36(4) has been labeled as C4 in the given phylogenetic tree.

Kathmandu valley (Nepal). Of 90 potential LAB isolates, 31 isolates were phenotypically identified to be *Lactobacillus* spp. fulfilling basic *in vitro* probiotics characteristics. Of these, at least 4 were genetically confirmed to be *Lactobacillus* spp. Further *in vivo* experiments need to be conducted to assess gut adaptation and probiotics performance to develop these isolates as poultry probiotics feed supplement. Industrial production parameters, such as ability to propagate in cheaper growth media, and maintain viability throughout production steps till storage and final application also need to be assessed [10].

With an increasing global threat of antimicrobial resistance (AMR), there is an urgent need to explore alternative solutions to antibiotic use. Routine use of sub-optimal dosages of antimicrobials in poultry sector with an aim to enhance their growth and performance has been one of the drivers of accelerated AMR [26]. The discontinuation of anti-microbial use in poultry, however, as a result of stringent regulation in high income countries, has resulted in poor poultry production [26]. This underscored the importance of prophylactics like antimicrobials for desired poultry yields. There are numerous studies indicating the use of probiotics as alternative solution replacing antimicrobials in poultry industry with similar benefits but without risk of AMR emergence [26]. Additionally, probiotics are also known to inhibit aflatoxigenic molds and degrade aflatoxins [26].

There are various native breeds of chicken found in Nepal, generally known as "local breeds", but some with specific names like Shakini, Giriraj, Ghantikhuile, Kadaknath and Pwankh Ulte (Dumse) [27]. Based on various factors, gut microbiota of free-ranging indigenous chicken breeds is different from that of commercial breeds [28]. Free-ranging chicken forage on a variety of natural food sources, such as insects and green foliage, which may enrich their microbiome diversity [29]. As a result, compared to commercial breeds, the backyard breeds have higher immunity, and therefore, can be more resilient against various infectious diseases [29].

We have isolated potential probiotics from indigenous chicken breeds. Unlike probiotics that are often isolated from abiotic sources, such as fermented food, the probiotics isolated in our study are host-specific to chicken with potential enhanced intestinal mucosa colonization abilities for greater natural integration into host gut microbiota [30]. However, further *in vivo* investigation is still needed to fully assess this probiotics feature.

Bacterial infections are one of the major reasons of morbidity and mortality in poultry, especially in commercial broiler and layer breeds- leading to production loss [31]. Colibacillosis and salpingitis caused by pathogenic strains of *E. coli*, mycoplasmosis caused by *Mycoplasma* spp., fowl cholera by *Pasteurella multocida*, necrotic enteritis by *Clostridium perfringens*, ulcerative enteritis by *Clostridium colinum*, salmonellosis, fowl typhoid and paratyphoid by diverse serovars of *Salmonella* spp., omphalitis by coliforms, pododermatitis and staphylococcosis by *Staphylococcus aureus*, and shigellosis by *Shigella* spp. are some of the common diseases caused by bacterial infections in poultry [31, 32].

Probiotics are known to protect host from infections through various mechanisms, including inhibiting colonization by pathogens and producing antibacterial metabolites such as acetic acid, lactic acid, alcohols, and bacteriocins [33]. Effective inhibition property against known pathogenic bacteria is one of the important selection criteria for good probiotics candidate. In our study, inhibition of coliform bacteria by LAB isolates was more common compared to inhibition against Gram-positive cocci, where 67% (31/46) phenotypic LAB isolates inhibited *Salmonella* spp., while 19% (9/46) isolates inhibited all tested enteric bacilli, in contrast to 15% (7/46) isolates that inhibited *S. aureus* with or without inhibiting enteric bacilli. Taheri *et. al* also reported higher inhibition of enteric bacilli, where 18% (62/332) of selected LAB strains isolated from gut of broiler chicken inhibited *Salmonella enteritidis*, *Salmonella typhimurium*, and *E. coli* O78:K80 [34]. Kizerwette-Swida *et al.* reported higher inhibition activity against Gram-positive pathogens including *C. perfringes* and *S. aureus* than coliforms including *E. coli* and *Salmonella* [35]. In our study, 5 of 46 LAB isolates inhibited all tested Gram-positive cocci and enteric bacilli, 3 (26B, 28B, and 30B) of which were genetically identified to be *L. plantarum*. Probiotics cultures of *L. plantarum* has been reported to have efficient antibacterial activity against broad spectrum of bacteria including clinical isolates *S. aureus* and *E. coli* via direct cell competitive exclusion as well as production of acids or bacteriocin-like inhibitors [36, 37].

The ability to survive in acidic gut environment, as they pass through stomach and intestine and colonize in the host gut, is an essential property of probiotics [30]. In our study, 27 of 46 isolates fulfilling phenotypic characteristics of LAB isolates showed survival tolerance in simulated gastric juice medium of pH 3.0, with the survival ratio ranging from 1.2% to 62.1%. This infers that the isolates would survive harsh acidic condition of the host gut upon introduction. Feng *et al.* reported that of 52 potential probiotics strains isolated from gut of piglets, 100% of isolates showed survival at pH 3.0, though 8 isolates failed to survive at an acidic pH of 1.0 [14]. In another study aiming to isolate potential probiotics from poultry fecal samples, of 42 potential isolates, 16% (n = 7) of selected isolates tolerated acidic pH of 2.5 and 4.0 compared to normal pH [38]. Further, lactic acid produced by LAB isolates provides various health benefits including immune-modulatory functions and prevention of diarrheal diseases [1]. In our study, 37 of 46 potential probiotics produced lactic acid with the yield ranging from 38 to 81%.

Bile salts are one of essential constituents of mammalian gut that help solubilize dietary fats. Hydrophobicity and detergent properties of bile salts exert strong antimicrobial effects. Thus, tolerance to bile is considered as one of important qualities of probiotics. Bile concentration varies along duodenum, jejunum and cecum of chicken gut, which were estimated to be 0.17, 0.7, and 0.0085% respectively [39]. In our study, 18 of phenotypic lactobacilli isolates fulfilling probiotics characteristics of acid production and acid tolerance showed 24 hour *in vitro*

survival in the presence of 0.3% and 0.5% bile salts. Shin *et al*. reported that all 3 sequenced potential probiotics isolates obtained from gut of broiler chicken tolerated the presence of 0.3% bile salts [40]. Oh *et al*. reported 7 of the potential probiotics isolates tolerated 0.3% bile salts for 24 hours [41].

## Conclusion

Our study isolated, characterized and identified 4 *Lactobacillus* spp. demonstrating optimal *in vitro* probiotics properties. With further evaluation, these identified *Lactobacillus* spp. candidates can be developed as effective probiotics which can be used as poultry feed supplement replacing regularly used antimicrobials.

## Supporting information

**S1 Table. Demographic details of chicken cloacal samples shown by identification number, sampling locations and the chicken breed.**
(DOCX)

**S2 Table. Laboratory results of 52 LAB isolates on antibacterial inhibition, biochemical test, and *in vitro* test for potential probiotics strains.**
(DOCX)

## Acknowledgments

We acknowledge the field and laboratory team of the Center for Molecular Dynamics Nepal (CMDN) for providing logistical, operational and supervisory support throughout the study. We thank the SANN International College for their support. And we express our gratitude to the members of local community where we collected our samples.

## Author Contributions

**Conceptualization:** Mohan Gupta, Dibesh Karmacharya.

**Data curation:** Sulochana Manandhar, Gaurab Karki, Pragun Rajbhandari, Prajwol Manandhar.

**Formal analysis:** Roji Raut, Sulochana Manandhar, Pragun Rajbhandari, Prajwol Manandhar.

**Funding acquisition:** Mohan Gupta, Dibesh Karmacharya.

**Investigation:** Mohan Gupta, Roji Raut, Ashok Chaudhary, Ujwal Shrestha, Saubhagya Dangol, Gaurab Karki.

**Methodology:** Mohan Gupta, Roji Raut, Ashok Chaudhary, Ujwal Shrestha, Saubhagya Dangol, Keshab Raj Budha, Rajindra Napit.

**Project administration:** Ujwal Shrestha, Sudarshan G. C., Rajindra Napit.

**Resources:** Sudarshan G. C., Rajindra Napit, Dibesh Karmacharya.

**Software:** Pragun Rajbhandari, Prajwol Manandhar.

**Supervision:** Ashok Chaudhary, Prajwol Manandhar, Rajindra Napit, Dibesh Karmacharya.

**Validation:** Sulochana Manandhar, Christian Gortazar, José de la Fuente, Pragun Rajbhandari, Prajwol Manandhar.

**Visualization:** Roji Raut, Sulochana Manandhar, Sudarshan G. C., Sandra Díaz-Sánchez, Christian Gortazar, José de la Fuente.

**Writing – original draft:** Mohan Gupta, Sulochana Manandhar.

**Writing – review & editing:** Mohan Gupta, Roji Raut, Sulochana Manandhar, Saubhagya Dangol, Keshab Raj Budha, Gaurab Karki, Sandra Díaz-Sánchez, Christian Gortazar, José de la Fuente, Dibesh Karmacharya.

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
