## [Decision Letter · Decision Letter 0]

8 Sep 2022

PONE-D-22-20489Identification and characterization of probiotics isolated from indigenous chicken (Gallus domesticus) of NepalPLOS ONE

Dear Dr. Karmacharya,

Thank you for submitting your manuscript to PLOS ONE. After careful consideration, we feel that it has merit but does not fully meet PLOS ONE’s publication criteria as it currently stands. Therefore, we invite you to submit a revised version of the manuscript that addresses the points raised during the review process.

I concur with all the comments raised by the reviewers. Please address all of the issues in the revised manuscript.

We look forward to receiving your revised manuscript.

Kind regards,

Xiaolun Sun

Academic Editor

PLOS ONE

Journal Requirements:

3. We note that Figure 1 in your submission contain map images which may be copyrighted. All PLOS content is published under the Creative Commons Attribution License (CC BY 4.0), which means that the manuscript, images, and Supporting Information files will be freely available online, and any third party is permitted to access, download, copy, distribute, and use these materials in any way, even commercially, with proper attribution. For these reasons, we cannot publish previously copyrighted maps or satellite images created using proprietary data, such as Google software (Google Maps, Street View, and Earth). For more information, see our copyright guidelines: http://journals.plos.org/plosone/s/licenses-and-copyright.

Reviewers' comments:

Reviewer's Responses to Questions

**Comments to the Author**

1. Is the manuscript technically sound, and do the data support the conclusions?

Reviewer #1: Partly

Reviewer #2: Yes

2. Has the statistical analysis been performed appropriately and rigorously? 

Reviewer #1: N/A

Reviewer #2: Yes

3. Have the authors made all data underlying the findings in their manuscript fully available?

Reviewer #1: Yes

Reviewer #2: Yes

4. Is the manuscript presented in an intelligible fashion and written in standard English?

Reviewer #1: No

Reviewer #2: Yes

5. Review Comments to the Author

Reviewer #1: In this manuscript, Mohan Gupta and his colleagues isolated, characterized, and identified 4 Lactobacillus spp. from indigenous chicken of Nepal demonstrating optimal probiotic activities in vitro, which may be used as antibiotic additive alternatives. However, lack of in vivo experiment and novel findings limit the integrality of this paper. Language modification is recommended. The other comments can be found below:

1. At line 4, please make sure your short title is “Isolation and identification of probiotics from native chicken breeds of Nepal” or “Isolation and identification of probiotics from chicken of Nepal”? Please keep consistent.

2. Please give references for sentences at line 76-77 and 77-78.

3. At line 107, 110, 120, 124, 190, 205, 206, 207, there should be spaces between words “with0.5%”, “India)with0.5%”, “O.D630adjustedbacterial”, “0.5O.D630”, “fromvarious”, “37produced”, “37isolates, 27isolates”, “17isolates”. I will not list all of those kind of error, please correct throughout the manuscript.

4. At line 88, there shouldn’t be a space before “Sampling”.

5. At lines 126-128, what’s the criterion you used to determine whether the LAB isolates have anti-bacterial activity or not? Although you mentioned in the results part at line 195 that “pathogen with zone of inhibition ranging from 7 to 16mm”, please give a reference, and is the same criterion (zone range) applied for every pathogen?

6. How these 90 potential LAB isolates distributed among the 41 cloacal samples? How do you distinguish the LAB isolates from same cloacal/plate sample?

7. At lines 199-200, please delete “of which 5 isolates also inhibited S. aureus.”, it’s repetitive.

8. According to your description from lines 136-146, please make clear at line 205, the 38%-81% is acid yield or percentage of acid yield.

9. At lines 151-153, to calculate the acid tolerance, why not use two pH for same medium, here you use gastric juice media and MRS.

10. From line 202-209, looks like not all the LAB isolates applied to all the screening methods, please clarify how many isolates were tested in each method. In an other words, are these tests applied sequentially?

11. At line 223, please change “a” to “as”.

12. According to the formula at line 167, how you get value above 100% for the bile tolerance data at Table S2? Also, is that reasonable that the tolerance increases as the bile concentration increase, for example, data for 22(3)?

13. Conclusion for C4/36(4), although it has high tolerance to bile, but one of the important characteristics of probiotics is the ability to inhibit pathogen growth, looks it doesn’t fit.

14. It’s not necessary to put sentences at lines 301-302 in conclusion part.

Reviewer #2: Gupta et al., have presented important findings on characterising on probiotics for countering the effect of gut pathogens and rising AMR related global issues. I found this article very well designed and experiments are well executed. i

6. PLOS authors have the option to publish the peer review history of their article (what does this mean?). If published, this will include your full peer review and any attached files.

Reviewer #1: No

Reviewer #2: No

---

## [Author Response · Author response to Decision Letter 0]

21 Sep 2022

Point by point response to editor’s comments:

- The file names have been corrected.

- Partial fund for conducting the laboratory experiments were received as academic thesis support from SANN International College, Purbanchal University (by the authors Mohan Gupta, Ujwal Shrestha, Saubhagya Dangol, and Sudarshan GC). Rest of the fund for conducting this study was obtained from institutions BIOVAC Nepal and Center for Molecular Dynamics Nepal (CMDN).

b) State what role the funders took in the study. 

- The funders had role in data analysis and preparation of the manuscript study design. 

- Roji Raut, Sulochana Manandhar, Ashok Chaudhary, Pragun Rajbhandari, and Prajwol Manandhar received salary from Center for Molecular Dynamics Nepal (CMDN).

- Rajindra Napit, and Dibesh Karmacharya received salary from BIOVAC Nepal.

3. Question on Figure 1:

- Figure 1 is not a copy right material. Instead, this has been created by the authors using the program Q-GIS (version 3.22.7) based on the recorded GPS locations. We permit the figure to be used freely by anyone for download, copy, distribute and any other way.

Response to the comments of Reviewer 1

1. At line 4, please make sure your short title is “Isolation and identification of probiotics from native chicken breeds of Nepal” or “Isolation and identification of probiotics from chicken of Nepal”? Please keep consistent.

- This has been corrected in the revised manuscript now by replacing with the former phrase.

2. Please give references for sentences at line 76-77 and 77-78.

- The reference to the first sentence was provided. For the second sentence, the use, and particularly production of probiotics in poultry and other livestock sectors in Nepal is relatively much less practiced. This sentence has been accordingly modified in the revised manuscript now.

3. At line 107, 110, 120, 124, 190, 205, 206, 207, there should be spaces between words “with0.5%”, “India) with 0.5%”, “O.D630adjustedbacterial”, “0.5O.D630”, “fromvarious”, “37produced”, “37isolates, 27isolates”, “17isolates”. I will not list all of those kind of error, please correct throughout the manuscript.

- We apologies for the typo. It seems like when the Microsoft word document is shared between authors who have different versions of operating system, the problem of word spacing automatically appears. We have checked all the text word by word in the revised manuscript now. Thank you for pointing this out.

4. At line 88, there shouldn’t be a space before “Sampling”.

- This has been corrected in the revised manuscript now.

5. At lines 126-128, what’s the criterion you used to determine whether the LAB isolates have anti-bacterial activity or not? Although you mentioned in the results part at line 195 that “pathogen with zone of inhibition ranging from 7 to 16mm”, please give a reference, and is the same criterion (zone range) applied for every pathogen?

- The production of clear zone of any size around the inoculated wells was taken as an indication of antibacterial activity imposed by that LAB isolate against the given pathogen inoculated as a lawn culture. This criterion has been applied for every tested pathogen. In this study, the observed zones of inhibition ranged between 7 mm to 16 mm. This has been clarified in the revised manuscript, and referenced too.

6. How these 90 potential LAB isolates distributed among the 41 cloacal samples? How do you distinguish the LAB isolates from same cloacal/plate sample?

- Multiple morphologically distinct colonies producing clear hydrolyzing zones and ranging from white to cream in colour and measuring 3-4 mm in size were selected as these could be potentially distinct LAB isolates. Because we used cloacal swab as source sample, it is expected to harbor diverse culturable microbiome, many of which could be potential LAB strains. This has been clarified in the revised manuscript now.

7. At lines 199-200, please delete “of which 5 isolates also inhibited S. aureus.”, it’s repetitive.

- This has been responded accordingly in the revised manuscript now.

8. According to your description from lines 136-146, please make clear at line 205, the 38%-81% is acid yield or percentage of acid yield.

- This has been clarified in the revised manuscript now.

9. At lines 151-153, to calculate the acid tolerance, why not use two pH for same medium, here you use gastric juice media and MRS.

- We wanted to test if the bacterial growth invitro in its favorable MRS medium was comparable to the actual gastric juice present invivo. The use of two pH for same medium would have been easier by simply adjusting the pH to 3 and 7, however with our long term motive to establish it as a feed supplement, we used gastric juice media for more realistic comparision.

10. From line 202-209, looks like not all the LAB isolates applied to all the screening methods, please clarify how many isolates were tested in each method. In an other words, are these tests applied sequentially.

- Yes, not all LAB isolates were subjected to all screening methods. Only those isolates testing positive in each screening step were tested further in the next following test. The order of the tests was as mentioned sequentially in the Methods section. That is, the exact sequence were the selective isolation in CaCO3 media (N=90), followed by antimicrobial susceptibility test (N=52), phenotypic test (Gram staining and biochemical tests)(N=46), and test for lactic acid production (N=37). Finally, these 37 isolates fulfilling all above screening parameters were subjected to tolerance tests to acid and bile salt.

The sequence of screening has been mentioned in methods. For further clarification, explanations have been added in the methods section under ‘Test for acid tolerance’ and ‘Test for bile tolerance’.

11. At line 223, please change “a” to “as”.

-This has been responded accordingly in the revised manuscript now.

12. According to the formula at line 167, how you get value above 100% for the bile tolerance data at Table S2? Also, is that reasonable that the tolerance increases as the bile concentration increase, for example, data for 22(3)?

- The formula for bile tolerance is based on absorbance defined by CFU. The value above 100% signifies, the bacterial number with bile has doubled faster than without bile. The increase in bile concentration doesn’t necessarily increase the bile tolerance, infact, less than 50% of the isolates showed no tolerance to bile with even lesser isolates able to tolerate 1% bile. A similar pattern was shown in the works by Anukam & Koyama, 2007 whereby the CFU increased upto 0.5% bile concentration and decreased on further increased bile cocncentration. Here, we selected the samples with highest tolerance at 0.3% and reduced bile tolerance at 0.5% and 1% bile concentration. The reason we did not select 22(3).

13. Conclusion for C4/36(4), although it has high tolerance to bile, but one of the important characteristics of probiotics is the ability to inhibit pathogen growth, looks it doesn’t fit.

- Although C4/36(4) doesn’t fit the most essential characteristic of pathogen growth, we have selected it for molecular screening. The isolate doesn’t seem to have any significant feature to be an isolate of choice for molecular screening as it showed no pathogen inhibition invitro. Still we have chosen it because, we are unaware of the behavior of the bacteria invivo , the actual mechanism by which the bacteria inhibits the pathogen. So, if the bacteria has a high enough tendency to tolerate bile, it may show a different behavior invivo with pathogen interaction; which is again still our hypothesis. So, having 3 isolates with all the desired characteristics, we have selected one entirely different isolate. The molecular screening indeed showed the isolate to be homologous to L.reuteri, one very potential probiotic candidate.

14. It’s not necessary to put sentences at lines 301-302 in conclusion part.

-This has been responded accordingly in the revised manuscript now.

---

## [Decision Letter · Decision Letter 1]

10 Nov 2022

PONE-D-22-20489R1Identification and characterization of probiotics isolated from indigenous chicken (Gallus domesticus) of NepalPLOS ONE

Dear Dr. Karmacharya,

Thank you for submitting your manuscript to PLOS ONE. After careful consideration, we feel that it has merit but does not fully meet PLOS ONE’s publication criteria as it currently stands. Therefore, we invite you to submit a revised version of the manuscript that addresses the points raised during the review process.

I concur with all the comments raised by the reviewers. please address these issues in the revised manuscript.

We look forward to receiving your revised manuscript.

Kind regards,

Xiaolun Sun

Academic Editor

PLOS ONE

Journal Requirements:

Reviewers' comments:

Reviewer's Responses to Questions

**Comments to the Author**

1. If the authors have adequately addressed your comments raised in a previous round of review and you feel that this manuscript is now acceptable for publication, you may indicate that here to bypass the “Comments to the Author” section, enter your conflict of interest statement in the “Confidential to Editor” section, and submit your "Accept" recommendation.

Reviewer #1: (No Response)

2. Is the manuscript technically sound, and do the data support the conclusions?

Reviewer #1: Partly

3. Has the statistical analysis been performed appropriately and rigorously? 

Reviewer #1: N/A

4. Have the authors made all data underlying the findings in their manuscript fully available?

Reviewer #1: Yes

5. Is the manuscript presented in an intelligible fashion and written in standard English?

Reviewer #1: (No Response)

6. Review Comments to the Author

Reviewers: In this revised manuscript, two main questions should be clarified. Firstly, please clarify the fundings, institutions, authors, and their roles in this paper, secondly, please make a clear explanation of the equations you used in this study, it’s better to give any references if someone used that before. I checked the references you cited, they only had word description, without results/equation explanation.

1. Multiple institutions were listed, but only three of them provided fundings. Please list the roles and responsibilities of institutions from France, USA, Spain, Australia played in this paper. Also, please list the contribution of each author.

2. At “Financial Disclosure” region, you claimed that “The funders had no role in study design, data collection and analysis, decision to publish, or preparation of the manuscript.” But when you answered the question 1b) of the editor, you claimed that “The funders had role in data analysis and preparation of the manuscript study design”. Which is the correct one.

3. The short title is essentially same as the title, please change it to a concise one.

4. At lines 78-80, the sentences should be rephrased as “However, the use and particular production of probiotics in poultry sectors in developing countries, like Nepal, is relatively less pronounced.”

5. At lines 140, the title was “Test for production of total lactic acid”, but the formula and results you demonstrated were percentage yield of lactic acid. Please make an explanation and provide the method you used to calculate the actual yield of lactic acid.

6. At lines 158-160, 168-170, different bacteria have different behavior, you need to make your own curve of the relationship of CFU/ml and OD value. And the reference you cited, looks like the calculation they did was wrong, according to their equation, OD1 was equal to 7.968*10^8 CFU/ml.

7. At line 162, there should be a space between “mlat”. And the equation should be Percent acid tolerance=(CFU/ml at pH3)/(CFU/ml at pH7), if you want to mean the percent of bacteria survive in acids. Same problem for the equation at line 171. If my understanding of these calculations is correct, you should adjust your results’ data.

8. What’s the reason not identify LAB first (Phenotypic bacterial identification), then do the analysis of antibacterial activity?

9. At line 199, from Table S2, looks zone of inhibition ranging was from 7 to 18 mm.

10. At lines 199-201, of the six pathogenic bacteria you tested, only Staphylococcus aureus is G positive, you can describe the results you had, but it’s unrepresentative.

11. At line 214, where did the number “31” come from? Since only 27 isolates from the 37 acid-producing isolates can survive in pH3.

12. Table S2, table note, please change CaCO3 to CaCO3.

7. PLOS authors have the option to publish the peer review history of their article (what does this mean?). If published, this will include your full peer review and any attached files.

Reviewer #1: No

---

## [Author Response · Author response to Decision Letter 1]

21 Dec 2022

In this revised manuscript, two main questions should be clarified. Firstly, please clarify the fundings, institutions, authors, and their roles in this paper.

Response: Funding: Partial fund for conducting the laboratory experiments were received as academic thesis support from SANN International College, Purbanchal University (by the authors Mohan Gupta, Ujwal Shrestha, Saubhagya Dangol, and Sudarshan GC). Rest of the logistical support for conducting this study was obtained from institutions BIOVAC Nepal and Center for Molecular Dynamics Nepal (CMDN). 

The authors, their affiliated institutions, and contribution in this study are already clearly mentioned in the PLOSone manuscript submission portal during paper submission.

 Secondly, please make a clear explanation of the equations you used in this study, it’s better to give any references if someone used that before. I checked the references you cited, they only had word description, without results/equation explanation.

Response: 

The equations used in the study have been explained in the manuscript and cited accordingly.

 Multiple institutions were listed, but only three of them provided fundings. Please list the roles and responsibilities of institutions from France, USA, Spain, Australia played in this paper. Also, please list the contribution of each author.

Response: All the other institutions provided intellectual and technical support in study design and manuscript preparation.

2. At “Financial Disclosure” region, you claimed that “The funders had no role in study design, data collection and analysis, decision to publish, or preparation of the manuscript.” But when you answered the question 1b) of the editor, you claimed that “The funders had role in data analysis and preparation of the manuscript study design”. Which is the correct one?

Response: The statement “The funders had role in data analysis and preparation of the manuscript and study design” is correct and amended accordingly in the paper.

3. The short title is essentially same as the title, please change it to a concise one.

Response: The short title has been replaced as “Characterization of probiotics from indigenous chicken” 

4. At lines 78-80, the sentences should be rephrased as “However, the use and particular production of probiotics in poultry sectors in developing countries, like Nepal, is relatively less pronounced.”

Response: This has been corrected accordingly.

5. At lines 140, the title was “Test for production of total lactic acid”, but the formula and results you demonstrated were percentage yield of lactic acid. Please make an explanation and provide the method you used to calculate the actual yield of lactic acid.

Response:

Actual yield of lactic acid was calculated using the formula, 

Actual lactic acid present in 100 ml of bacterial broth = percent yield of lactic acid = (Vg × N × 90 × 100)/Vm, 

Where, Vg= volume of NaOH solution added for titration

N= concentration of sodium hydroxide standardized solution expressed in Eq/L

90= equivalent weight of lactic acid

Vm= volume of LAB broth culture used for titration

As gram equivalent weight of NaOH is 40, and we used 0.1 N of NaOH,

And, balanced chemical equation for reaction between NaOH and Lactic acid is as:

C3H6O3 + NaOH → NaC3H5O3 + H2O

ie. 1 mole of NaOH requires 1 mole of Lactic acid for neutralization.

Since, 1 mole of NaOH = 40gm and 1 mole of Lactic acid = 90.08g

40gm NaOH requires 90.08g Lactic acid

Ie. 1 gm of NaOH requires 2.25 gm Lactic acid

Thus, actual lactic acid yield = (Vg X 4 X 2.25)/1000 x Vm

Here, Vm= 2 as we used 2 ml of LAB culture for initial titration.

Thus, actual lactic acid yield = (Vg X 4 X 2.25)/1000 x 2

 = Vg X 0.0045

For example, for our sample 1 (1), volume of NaOH required for titration was 2.5 ml. 

Thus, % lactic acid yield for this sample 1 (1) = actual yield/theoretical yield X 100

% lactic acid yield for sample 1 (1) = (2.5 X 0.0045) X 100 = 56.25

 0.02

This formula has been detailed in the article “Fabro M, Milanesio H, Robert L, Speranza J, Murphy M, Rodríguez G, et al. Determination of acidity in whole raw milk: comparison of results obtained by two different analytical methods. Journal of dairy science. 2006;89(3):859-61.” which we have cited in the paper.

6. At lines 158-160, 168-170, different bacteria have different behavior, you need to make your own curve of the relationship of CFU/ml and OD value. And the reference you cited, looks like the calculation they did was wrong, according to their equation, OD1 was equal to 7.968*10^8 CFU/ml.

Response: 

As per the suggestion, we did perform the growth curve for one of our bacterial isolate 28B (1) that is Lactobacillus plantarum, and obtained the growth curve as shown in Figure 1. 

Figure 1. Growth curve analysis of isolate 28B(1) 

Further, we established the OD and CFU relationship as shown in Figure 2. We obtained the equation for relationship as y=7.3249+4.3782.

Figure 2. Correlation between CFU and OD

Using this equation, we could obtain similar percentage increase in bile (table 1) and acid tolerance (table 2) as we obtained using the equation from the paper (Trabelsi et.al. 2013).

Table 1. Comparison of bile tolerance using two equations

Since no larger variation was observed in our final data, the OD and CFU correlation seems to be consistent irrespective of the reading. 

7. At line 162, there should be a space between “mlat”. And the equation should be Percent acid tolerance=(CFU/ml at pH3)/(CFU/ml at pH7), if you want to mean the percent of bacteria survive in acids. 

Same problem for the equation at line 171. If my understanding of these calculations is correct, you should adjust your results’ data.

Response: 

We calculated percent acid tolerance as (CFU/ml at pH 7.0-CFU/ml at pH 3.0)/(CFU/ml at pH 7.0) x100% as calculated in a cited paper “Farid W, Masud T, Sohail A, Naqvi S, Qazalbash MA. Molecular characterization and 16S rRNA sequence analysis of probiotic lactobacillus acidophilus isolated from indigenous Dahi (Yoghurt). International Journal of Bioscience. 2016;9(5):19-27” 

8. What’s the reason not identify LAB first (Phenotypic bacterial identification), then do the analysis of antibacterial activity?

Response: We thank the reviewer for the suggestion, which is in fact a standard approach. We, on the other hand, adopted a reverse approach where we first assessed the antibacterial activity followed by phenotypic bacterial identification. It was because we wanted to isolate potential LAB having antibacterial activity. In earlier pilot study, we found that large proportion of phenotypically identified LAB isolates lacked antibacterial activity. Thus, to reduce our work load and resources, we first performed antibacterial assessment of potential LAB isolates, and only those producing inhibitory activity were subjected to phenotypic bacterial identification.

9. At line 199, from Table S2, looks zone of inhibition ranging was from 7 to 18 mm.

Response: This has been corrected now.

10. At lines 199-201, of the six pathogenic bacteria you tested, only Staphylococcus aureus is G positive, you can describe the results you had, but it’s unrepresentative.

Response: Yes, we agree on this. This sentence has been removed now.

11. At line 214, where did the number “31” come from? Since only 27 isolates from the 37 acid-producing isolates can survive in pH3.

Response: This has been corrected now.

12. Table S2, table note, please change CaCO3 to CaCO3.

Response: This has been corrected now.

---

## [Editor Report · Decision Letter 2]

29 Dec 2022

Identification and characterization of probiotics isolated from indigenous chicken (Gallus domesticus) of Nepal

PONE-D-22-20489R2

Dear Dr. Karmacharya,

We’re pleased to inform you that your manuscript has been judged scientifically suitable for publication and will be formally accepted for publication once it meets all outstanding technical requirements.

Kind regards,

Xiaolun Sun, PhD

Academic Editor

PLOS ONE

---

## [Editor Report · Acceptance letter]

5 Jan 2023

PONE-D-22-20489R2 

Identification and characterization of probiotics isolated from indigenous chicken (*Gallus domesticus*) of Nepal 

Dear Dr. Karmacharya:

I'm pleased to inform you that your manuscript has been deemed suitable for publication in PLOS ONE. Congratulations! Your manuscript is now with our production department. 

Kind regards, 

on behalf of

Dr. Xiaolun Sun 

Academic Editor

PLOS ONE